



# Sea Ice Concentration Estimates from ICESat-2 Linear Ice Fraction. Part 1: Multi-sensor Comparison of Sea Ice Concentration Products

Ellen M. Buckley[1], Christopher Horvat[2], and Pittayuth Yoosiri[3]

[1]Department of Earth Science and Environmental Change, University of Illinois Urbana-Champaign, Champaign, IL, USA
[2]Department of Earth, Environmental, and Planetary Sciences, Brown University, Providence, RI, USA
[3]Department of Atmospheric and Oceanic Science, University of Maryland, College Park, MD, USA

**Correspondence:** Ellen Buckley (buckley0@illinois.edu)

**Abstract.** Sea ice coverage is a key indicator of changes in polar and global climate. Observational estimates of the area and extent of sea ice are primarily derived from passive microwave surface emissions, which are used to develop gridded products of sea ice concentration (SIC). Passive microwave (PM) satellite sensors remain the sole global product for understanding SIC variability. Here, in Part I of a two-part study, we use a dataset of more than 70,000 high-resolution airborne optical classified

images from Operation IceBridge, and we first identify biases in commonly used passive microwave products in areas with thin sea ice fractures. We find that passive microwave-derived SIC products overestimate true SIC with biases on average 4.4% in winter and 3.2% in summer. We show that ICESat-2, a laser altimeter operational since 2018, has the capacity to sample these thin fractures, with good agreement between ICESat-2 surface-type classifications and near-coincident Worldview and Sentinel-2 data in winter. Using the ICESat-2 surface type classifications, we introduce a new derived product, the linear ice

fraction (LIF) and discuss its potential for representing a two-dimensional sea ice concentration field. This paper highlights the biases present in PM-derived SIC and makes a case for considering the integration of ICESat-2 and its high-precision measurements of the sea ice surface to enhance future SIC estimations. In Part II, we identify and evaluate biases associated with the development of a gridded LIF product and compare it to existing PM-SIC data.

## 1   Introduction

Sea ice concentration (SIC), the fraction of an ocean area covered by sea ice, is critically important for understanding Earth's climate variability. Since the late 1970s, SIC is estimated globally and daily using passive microwave (PM) satellites at both hemispheres. Numerous algorithms (at least 11, (Kern et al., 2019)) have been developed which convert surface radiative properties into gridded SIC on time scales from days to months. PM-derived SIC is a standard for assessing sea ice state and change (Meredith et al., 2022), and is assimilated into state-of-the-art forecast and climate models at both hemispheres

(Mazloff et al., 2010; Sakov et al., 2012; Massonnet et al., 2015; Verdy and Mazloff, 2017; Zhang et al., 2018; Fritzner et al., 2019; Zhang et al., 2021). Yet these SIC products are constrained by various shortcomings of PM sensors, including their coarse resolution and sensitivities to surface water, which prevent them from accurately capturing small-scale features and certain sea ice properties (Ivanova et al., 2015; Kern et al., 2016).





Sea ice is a heterogeneous, fractured mosaic of solid floes or plates ranging in size from meters to hundreds of kilometers
and whose surface is comprised of some combination of ice, snow, and meltwater. Cracks in the sea ice, known as leads, are
narrow in width and vary over length scales of kilometers to hundreds of kilometers and open and close on timescales of
minutes to weeks (Bouillon and Rampal, 2015; Hutter et al., 2019; Olason et al., 2020; Hutter and Losch, 2020). Uncertainty
in PM-derived SIC can arise from the presence of leads, which are challenging to detect due to their near-linear geometry and
high variability. When examining 11 different PM SIC products in regions with near-100% SIC in winter, Kern et al. (2019)
found systematic algorithmic differences between products that range from -1.1% to 3.5%. While these differences are small
in terms of the overall SIC, air-sea exchange in leads is an important source of ocean mixing and energy in winter. A second,
larger discrepancy in PM-SIC comes in summer, when PM-SIC estimates vary up to 35% (Kern et al., 2020). Melt ponds on
the sea ice appear radiometrically similar to open water, and can be conflated with open water (Ulaby et al., 1986; Kern et al.,
2016), hampering the ability of PM algorithms to ascertain the true sea ice coverage.

Local errors in PM-SIC are observed to have a compensating effect when integrated over the Arctic or Antarctic, and
hence the impact of algorithmic uncertainty or bias on estimates of total sea ice coverage is estimated to be less than 1%,
even in summer (Notz, 2015; Meier and Stewart, 2019; Kern et al., 2020). Still, no independent alternative to PM exists for
measuring SIC from local to global scales. Thus it is not clear whether biases exist in PM-SIC algorithms that go beyond
normally-distributed uncertainties, which might affect climate process understanding, forecast model data assimilation, and
future projections.

In this, study, we investigate an independent measure of sea ice presence, the linear ice fraction (LIF), developed using
NASA's ICESat-2 laser altimeter (IS2). IS2 is a photon-counting laser altimeter with 0.7 m along-track sampling, a 10-meter
footprint, and high skill in differentiating sea ice and open water in non-summer months (Farrell et al., 2020; Kwok et al.,
2020, 2021). Compared to radar altimeters, IS2 is less susceptible to "snagging" by leads or melt ponds. IS2 can resolve Arctic
leads at the meter scale (Petty et al., 2021; Kwok et al., 2021), especially in winter, but has shown a limited ability to identify
melt ponds atop Arctic sea ice in summer (Farrell et al., 2020; Tilling et al., 2020). Importantly, IS2 (an active 532nm green
laser) does not rely on the PM signature of sea ice (radiation in the 10-100Ghz range) and has independent uncertainties.

We first explore errors and uncertainty in PM-SIC measurements using a set of more than 70,000 classified images from
NASA's Operation IceBridge Digital Mapping System (Buckley et al., 2020) in Sec. 3, illustrating the need to improve estimates
of SIC in compact and ponded ice. Using high-resolution imagery from different sources, we show that LIF derived from a
single ICESat-2 pass is at least as skilled at PM products at reconstructing local SIC for SIC near 100%. In Part II Horvat
et al. (2024), we construct uncertainty estimates for unsupervised LIF retrievals in the Arctic using these classified images,
which are explicitly constrained to build a global product. We then evaluate global differences between monthly IS2 LIF and
six commonly-used PM-SIC products at different resolutions.





**Table 1.** Passive Microwave Product Details

| Product | Product Abbr. | Sensor | Algorithm | Grid[a] | Dates Available | Reference |
|---|---|---|---|---|---|---|
| Sea Ice Concentrations from Nimbus-7 SMMR and DMSP SSM/I-SSMIS Passive Microwave Data, Version 2 | NT | SMMR, SSM/I-SSMIS | NASA Team | 25 km ps | 1Nov-78 - present | Cavalieri et al. (1984) |
| Bootstrap Sea Ice Concentrations from Nimbus-7 SMMR and DMSP SSM/I-SSMIS, Version 4 | BT | SMMR, SSM/I-SSMIS | NASA Bootstrap | 25 km ps | 1Nov78 - present | Comiso and Sullivan (1986) |
| NSIDC Climate Data Record | NSIDC | SMMR, SSM/I-SSMIS | NASA Team & Bootstrap | 25 km ps | 25Oct78 - present | Meier et al. (2014); Peng et al. (2013) |
| OSISAF Global Sea Ice Concentration Climate Data Record OSI-430-A OSI-450-A | OSI | SSMIS | OSI | 25 km EASE2 | 24Oct78 - 31Dec20 (405) 1Jan21 - present (403) | Lavergne et al. (2019) |
| AMSR-E/AMSR2 Unified L3 Daily 25 km Brightness Temperatures & Sea Ice Concentration Polar Grids, Version 1 | AMSR2-NT | AMSR2 | NASA Team 2 | 25 km ps | 2Jul12 -present | Meier et al. (2018) |
| Sea Ice Concentration data from AMSR-E, AMSR2 & SSMIS, U Bremen and U Hamburg ASI algorithm, Version 1 | AMSR2-ASI | AMSR-E/2, SSMIS | ARTIST | 6.25 km ps | 1Jun02 -present | Spreen et al. (2008a) |

[a]ps= polar stereographic

## 2 Passive Microwave Sea Ice Concentration Products

55

Passive microwave datasets provide long term, consistent information for the characterization of sea ice presence and variability. The launch of the scanning multichannel microwave radiometer (SMMR) in 1978 began the record of multichannel data that allowed for understanding surface types and ice temperature. Passive microwave sensors offer the advantage of being able to penetrate cloud cover and operate in both daylight and darkness, enabling year-round monitoring in the polar regions.

Brightness temperature can be used to calculate SIC and, in turn, sea ice area and extent. In this work, we utilize several data products from various sensors and algorithms to accurately represent the commonly used PM-SIC products within the modeling and observational community (Table 1).



## 2.1 Instruments

The Scanning Multichannel Microwave Radiometer (SMMR) was a 10-channel radiometer with both horizontal and vertical polarizations (Gloersen and Barath, 1977), which flew on the Nimbus-7 and SeaSat satellites. Although SeaSat was only operational for a few months, the SMMR instrument on Nimbus provided operations from 1978 to 1987. Satellites belonging to the Defense Meteorological Satellite Program (DMSP), a United States Department of Defense program, have carried the Special Sensor Microwave/Imager (SSM/I) radiometers and its successor, the Special Sensor Microwave Imager/Sounder (SSMI/S) radiometers. SSM/I radiometers were first launched on DMSP satellites from 1987 to 1999, and SSMIS radiometers from 2003 to 2014. The SSMIS is a 24-channel instrument that combines the imaging capabilities of the SSM/I with the Special Sensor Microwave Temperature sounder (Kunkee et al., 2008). The Advanced Microwave Scanning Radiometer (AMSR) series started with AMSR-J on JAXA's ADEOS-II launched in 2002, and was followed by AMSR-2 (NASA's Aqua, 2002) and AMSR-E (JAXA's GCOM WI, 2012) (Imaoka et al., 2010). Each advancement in passive microwave radiometers brings additional channels across a broader frequency spectrum, along with improved coverage and higher resolution.

## 2.2 Algorithms

Algorithms for converting brightness temperatures into sea ice concentration have also advanced, incorporating new spectral frequencies and addressing known data biases. Various studies have detailed the evolution, discrepancies, and limitations of these algorithms. We do not aim to provide an in-depth description of the previous work here. Instead, we offer a summary of the details relevant to our study. For more comprehensive information, readers are encouraged to consult the references provided in this section and above.

### 2.2.1 NASA Team

The NASA Team algorithm (NT, Cavalieri et al., 1984) was initially developed to determine sea ice concentration from SMMR and later modified for SSMI/s (Cavalieri et al., 1991). The observed brightness temperature is considered the sum of three surface types: open water, first year ice, and multiyear ice. Based on the differences in emissivities of these surface types at the frequencies captured by these sensors, radiance ratios are developed to distinguished the observed surface types. Radiance ratios are useful in that they are not dependent on the ice temperature variations but rather the relationship between ice temperature at different frequencies. This algorithm utilizes the 37 GHz vertical and the 19 GHz vertical and horizontal channels (37V, 19V, and 19H, respectively), and includes a weather filter based on two gradient ratios, one with 37V and 19V, and an additional one using the 22V and 19V channels (for SSMI/S-derived products). The NASA Team 2 algorithm Markus and Cavalieri (2009) was developed to address the sensitivity to emmissivity variations, specifically, low SIC bias in areas with deep snow. NASA Team 2 utilizes channels at higher frequencies (85 GHz for SSMI/S and 89 for AMSR) that feed into a radiative tranfer model to provide atmospheric correction to the retrievals.



### 2.2.2 NASA Bootstrap

The NASA Bootstrap algorithm (BT, Comiso and Sullivan, 1986) uses the distribution of brightness temperatures from 37H,
37V and 19V channels to determine surface types. Unlike the NASA Team algorithm, Bootstrap uses daily-varying tie points
to account for changing in surface conditions (e.g. melting). This algorithm also makes use of the 22V channel for reducing
atmospheric effects, and depends on an assumption in this algorithm the there exist regions in the Arctic that contain 100% ice
concentration.

Comiso et al. (1997) reports the biggest discrepancies between Bootstrap and observations in the marginal seas. In the Fram
Strait and the Northern Barents Sea, they found the BT algorithm produced SIC $\sim 5\%$ greater than those produced by NT -
though both, when compared with airborne SAR data, underestimate SIC at the edge of the ice pack. Validation efforts in the
Beaufort and Bering Seas conducted in 1998 found that the algorithms underestimate SIC compared to Landsat imagery by 8.2
% and 6.1 % on average for NT and BT, respectively (Cavalieri et al., 1991). The differences in SIC are caused by temperature,
emmissivity and tie point effects on the two algorithms, and both algorithms struggle with the identification of new, thin ice.

### 2.2.3 NSIDC Climate Data Record

To reduce biases for climate applications, the NSIDC Climate Data Record (NISDC) is a rule-based merge of the BT and
NT algorithms. The ice edge is defined where the BT algorithm finds SIC < 10%. Otherwise, the value of the NSIDC is the
maximum of the BT and NT algorithms. As both algorithms tend to underestimate SIC in different areas, this maximization
decreases the overall low bias in PM-SIC.

### 2.2.4 ARTIST Sea Ice

The Arctic Radiation and Turbulence Interaction Study (ARTIST) sea ice algorithm (ASI) from the University of Bremen and
University of Hamburg uses higher frequency channels (89V and 89H) in ASMR-E and AMSR-2 (Spreen et al., 2008b), which
allows for 4 times higher resolution products; these products are on a 6.25 km polar sterographic grid. The higher frequency
channel is more sensitive to weather, and thus a lower frequency channel is used for weather corrections. This algorithm also
utilizes fixed tie points which may lead to biases over seasonal changes or as the instrument degrade. It has been found that
the ASI algorithm, as well as the NT and BT, tend to underestimate sea ice concentration in the marginal ice zone (Alekseeva
et al., 2019).

### 2.2.5 Ocean and Sea Ice Satellite Applications Facility

The Ocean and Sea Ice Satellite Applications Facility (OSI SAF) CDR is a hybrid approach like the NSIDC-CDR, which
utilizing distinct methods for low SIC areas ($\leq 40$ % SIC), and high SIC areas (Lavergne et al., 2019; Tonboe et al., 2016). The
low SIC algorithm is derived from the BT algorithm, while the high SIC algorithm is based on the Bristol algorithm, which
incorporates polarization and spetral gradient information from the 19V, 37V, and 37H channels. Both employ dynamic tie
points and atmosphere correction and provide uncertainty estimates.



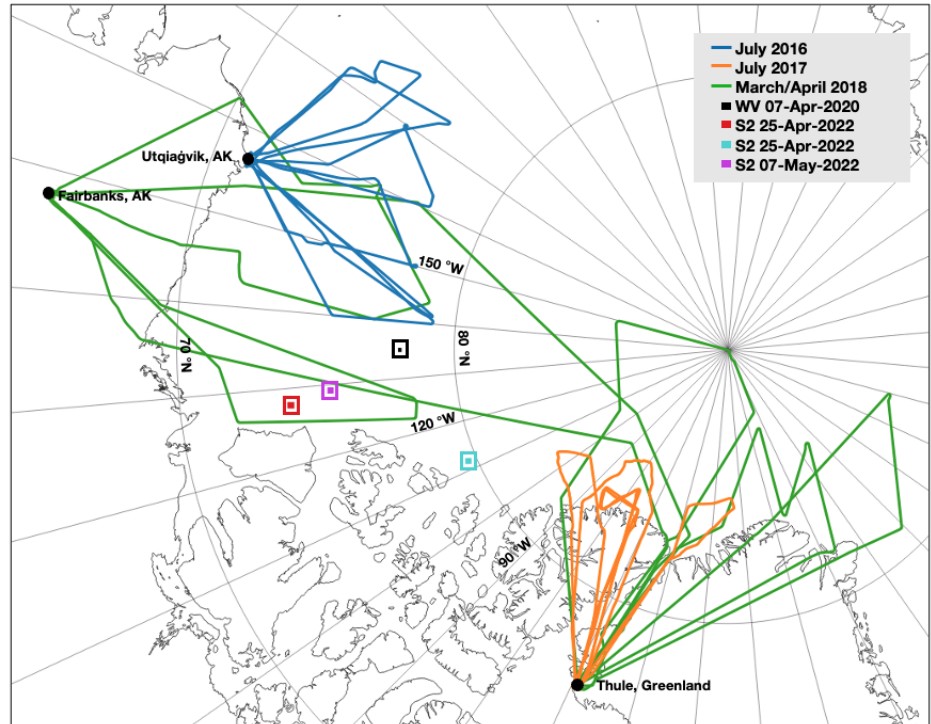

**Figure 1. Imagery Location.** Operation IceBridge flight lines for the July 2016 (blue) and 2017 (orange) summer Arctic sea ice campaigns, and the Spring 2018 flight lines (green). The footprints (boxed) of the WorldView and Sentinel-2 images used for validation of the ICESat-2 LIF: April 7, 2020 (black), April 25, 2022 (red), April 25, 2022 (cyan), May 7, 2022 (purple).

## 3 Comparing Sea Ice Concentration Products to High Resolution Visible Imagery

Operation IceBridge was a multi-year observational campaign which bridged the time period between the ICESat and ICESat-2 satellite operational eras. IceBridge flights captured along-track optical imagery of the sea ice surface - here we examine a set of 70,165 geolocated and orthorectified images taken in March and April 2018 (pre-surface melt), and during the July Arctic campaigns in 2016 and 2017 (during surface melt) (Dominguez, 2010). The Digital Mapping System (DMS) imagery has 0.1 m resolution and are approximately 400 m x 600 m. Each image is then processed according to the classification scheme of

Buckley et al. (2020) (hereafter B20), which classifies each image pixel into ice, open water and seasonal specific categories: melt pond in the summer, and new ice in the winter. Classified imagery was then visually validated. Details on the classification algorithm are available at (Buckley et al., 2020) and (Buckley et al., 2023).

Each B20-classified DMS image scene is compared to local SIC evaluated using six commonly used daily gridded PM-SIC 1. Since PM swaths (O 10 km) and image sizes (O 1 km) are not similar, we use two methods for comparing airborne point

measurements to the gridded satellite products. In the first method, we average all OIB SIC values inside of a single PM grid to account for varying ice conditions within the PM swath. In the second method, we take the center latitude and longitude of the





optical image and identify the grid cell in which this coordinate falls within the native grid of the SIC product. We find similar results, and we focus primarily on the first method (averaging DMS image statistics) as this is more representative of the entire PM cell. The results from method two can be found in Appendix A.

We only examine DMS images where all PM-SIC products have a SIC value above 15%, to avoid measurements outside the marginal ice zone. Since this study focuses on regions with sea ice leads, we limit our examination to winter locations where OIB imagery indicates SIC $\leq$ 99%. In the summer we examine images where MPF $\leq$ 50% to avoid outliers and misclassified images in the unsupervised analysis. When averaging OIB images to the PM grid scale, we are left with 20,498 unique OIB scenes: 15,270 points of comparison in "summer" and 5,228 in "winter". Comparative results are presented in Figure 2, with

winter results in the first column, summer results in the second, and the first method displayed in the top row, while the second method is shown in Appendix A.

Comparative statistics for all data are collected in Table 2, which include the mean SIC value ($\overline{\text{SIC}}$) and differences between the B20-derived SIC and the PM-derived SIC. The distribution of differences from the B20 data ($\Delta$) are then shown in Figure 2a and  2b, along with median differences and interquartile ranges (box).

**Table 2.** Comparison of mean and median differences in sea ice concentration for winter and summer.

|  | Winter (March-April 2018) | | | Summer (July 2016,2017) | | |
|---|---|---|---|---|---|---|
|  | B20 SIC Mean (%) | Mean $\Delta$ | Median $\Delta$ | B20 SIC Mean (%) | Mean $\Delta$ | Median $\Delta$ |
| NSIDC | 93.8 | 4.6 | 2.3 | 80.1 | 8.1 | 7.3 |
| BT | 93.8 | 4.5 | 2.3 | 80.1 | 8.2 | 7.3 |
| NT | 93.8 | 1.1 | 1.3 | 80.1 | -8.5 | -8.4 |
| AMSR | 93.8 | 4.9 | 2.1 | 80.1 | 10.0 | 9.2 |
| OSI | 93.6 | 3.5 | 0.5 | 84.2 | 1.4 | 4.4 |
| ASI | 91.6 | 5.5 | 0.8 | 80.6 | 1.0 | 3.9 |

## 150 3.1 Winter Sea Ice Concentration

During winter, in areas that are not 100% ice-covered (OIB SIC $\leq$ 0.99), PM products consistently overestimate SIC. When the OIB imagery-derived SIC is averaged within a PM grid cell, these products exhibit median positive biases ranging from 0.5% to 2.3%, with mean biases spanning from 1.1% to 5.5% (2). While this reflects a generally good agreement between the PM products and the SIC from imagery, even a small difference between 98% and 99% SIC can result in a doubling of the

155 open water area, significantly increasing heat and moisture exchange between the ocean and atmosphere. Median biases are lower than the mean biases, indicating that there are incidences of high overestimations of SIC in the PM datasets that impact the mean biases. The NSIDC product takes the higher of the BT and NT products, and because NT is consistently lower than BT, NSIDC biases display strong similarity in patterns to the BT biases.

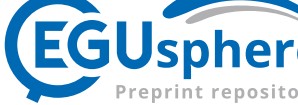

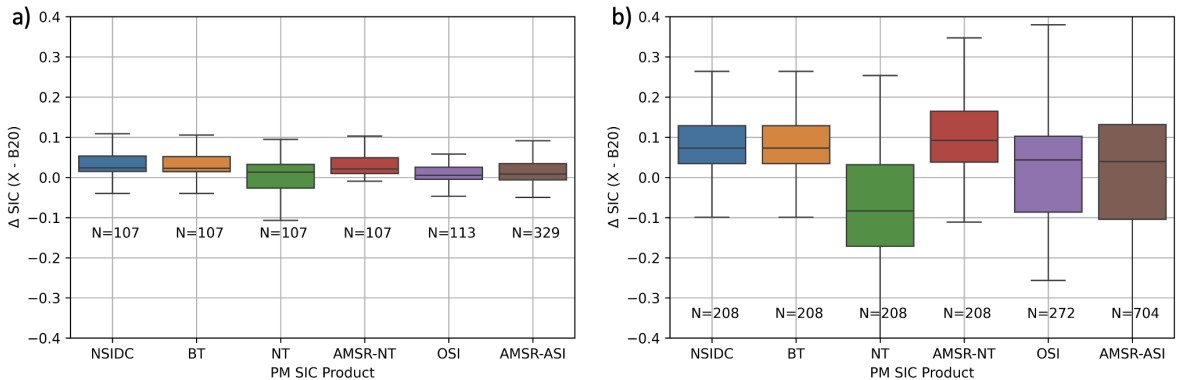

**Figure 2. Differences between Passive Microwave SIC retrievals and Operation IceBridge SIC for winter (a) and summer (b).** $\Delta$ SIC is given as the PM product less the OIB SIC value where values $> 0$ indicate the PM SIC product is great than imagery-derived SIC. OIB images SIC values within a PM grid cell are averaged for comparison with the PM SIC value. Winter scenes are included where OIB SIC $\leq$ 99%. Each boxplot shows the interquartile range (IQR) which is 25th percentile to 75th percentile. The line inside the boxplot represents the median. The whiskers show the range which here is defined as 1.5 times the IQR.

## 3.2 Summer Sea Ice Concentration

During summer, all PM products, except the NASA Team Algorithm using SSMIS data, exhibit a positive SIC bias. Median biases range from -8.4% to 9.2%, while mean biases range from -8.5% to 10%. Compared to winter biases, the summer absolute biases are greater and have a much wider range, indicating more uncertainty. We find that the NASA Team product provides the lowest values of SIC compared to the other algorithms and this trend is exacerbated in the summer. This is consistent with findings in Kern et al. (2020) which found NASA Team products estimate less SIC than other products in the Arctic in the

summer. Kern et al. (2020) found the NT algorithm products have a negative bias in the summer, as NT is known to be very sensitive to surface melt with fixed hemispheric tie points. They found that out of all the PM data tested, the greatest biases are in the groups including the BT, AMSR-NT, and NSIDC, with a positive bias of 5%–10%. They found the OSI-SAF product has the lowest absolute bias which is consistent with our findings (Figure 2b).

We grouped cell-averaged OIB melt pond fraction (MPF) derived from B20 into 10% intervals and analyzed the mean

absolute bias for each group. As shown in Figure 3, the absolute bias increases across all products as the melt pond fraction increases. This trend is most extreme with the NT product, indicating the melting sea ice surface strongly affects the algorithm accuracy. Interestingly, the AMSR product with the NT2 algorithm, has the second largest biases at high MPF values (MPF $\geq$ 25 %).

The comparison here is subject to important limitations, including uncertainties in surface classification and mismatches

between satellite footprints. We discuss the applicability and limitations of this approach in more detail in Sec. 5. Yet because

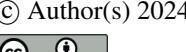



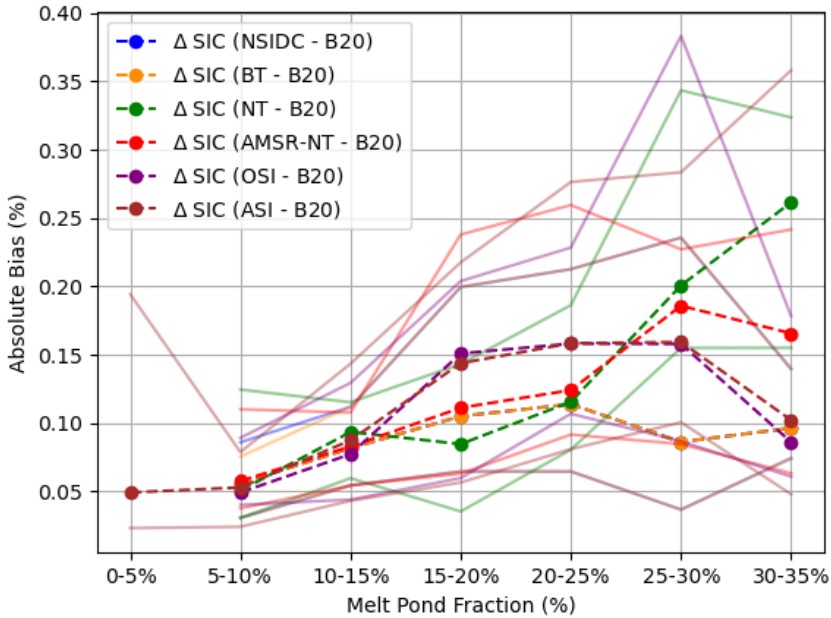

**Figure 3. Bias as a function of MPF.** Differences between Passive Microwave SIC retrievals and Operation IceBridge SIC in Summer v. Melt Pond Fraction. Biases are plotted at 5 % bins. The 25th and 75th percentile values are shown as faded lines.

of these consistent biases, we seek then to understand the applicability of alternative methods for retrieving SIC to reduce, understand, or constrain these uncertainties.

## 4 ICESat-2 and the Linear Ice Fraction

The Ice Cloud and land Elevation Satellite - 2 (ICESat-2; IS2) was launched in September 2018 carrying the Advanced To-
180 pographic Laser Altimeter System (ATLAS). The photon counting, 532 nm (green), six-beam laser system was designed specifically to measure height of the cryosphere and understand sea ice thickness distribution and elevation changes in ice sheets and glaciers (Markus et al., 2017). ICESat-2 allows for high-resolution sampling of the ice with a $\sim$ 11 m footprint Magruder et al. (2020), compared to the $\sim$ 70 m footprint of ICESat (Kwok et al., 2004). Although the six-beam configuration was developed to understand slope changes on ice sheets, it provides additional opportunities for observing the sea ice surface.
The high resolution and increased sampling has allowed for the resolution of narrow leads in the sea ice pack, that are crucial for determining accurate freeboard estimates. Kwok et al. (2019b) found that IS2 can consistently resolve leads as narrow as 27 m - though because of the angle between IS2 incidence and leads, finer crack features are likely recorded in IS2 sea ice products Hell and Horvat (2024).



**Table 3.** Comparison of LIF and SIC from ICESat-2, Drift Corrected Optical Imagery, and Passive Microwave Products.

| Date | | 07Apr2020 | | 25Apr2022 | | 27Apr22 | | 07May2022 | | |
|---|---|---|---|---|---|---|---|---|---|---|
| Location | | (-135,78.0) | | (-127.7,73.9) | | (-111.8,79.7) | | (-129.2, 75.4) | | |
| SIC | | SIC | $\overline{\Delta}$ | SIC | $\overline{\Delta}$ | SIC | $\Delta$ | SIC | $\Delta$ | $|\Delta|$ |
| **Optical** | | 97.5 | ∅ | 97.9 | ∅ | 98.7 | ∅ | 90.5 | ∅ | ∅ |
| **IS2** | Best | 97.5 | 0.0 | 95.7 | -2.1 | 98.7 | 0 | 92.4 | 1.9 | **1.0** |
| | ATL07 | 99.3 | 1.8 | 98.3 | 0.5 | 99.8 | 1.1 | 96.6 | 6.1 | **2.4** |
| **PM** | ASI | 94.5 | -3.0 | 98.7 | 0.9 | 97.0 | -1.7 | 96.7 | 6.1 | **2.9** |
| | OSI | 99.9 | 2.4 | 95.5 | -2.4 | 95.0 | -3.8 | 100.0 | 9.4 | **4.5** |
| | NSIDC | 100.0 | 2.5 | 100.0 | 2.1 | 100.0 | 1.3 | 100.0 | 9.5 | **3.8** |
| | NT | 100.0 | 2.5 | 100.0 | 2.1 | 100.0 | 1.3 | 100.0 | 9.5 | **3.8** |
| | BT | 99.2 | 1.7 | 100.0 | 2.1 | 100.0 | 1.3 | 100.0 | 9.5 | **3.6** |

The IS2 data set ATL07 consists of a set of along-track surface segments (Kwok et al., 2022). Each ATL07 segment is
created from an aggregate of 250 photons, with lengths ranging from ∼ 10 to 200 m in length depending on how reflective the
surface is (Kwok et al., 2019b). Segments are provided in locations where the local daily NSIDC-CDR sea ice concentration
exceeds 15%, and average ∼15 m for the strong beam and ∼60 m for the weak beam (Kwok et al., 2019a). A classification
algorithm is applied to the ATL07 segments to determine the surface contained in each segment. The goal of the classification
is to identify sea ice segments that can be used for freeboard (ATL10) calculation which requires measurements of the ice and
sea surfaces. The surface type classification parameter (seg_surf_type) is based on three parameters: surface photon rate, the
width of photon distribution and the background rate normalized to the sun elevation. This results in five main classification
categories and associated parameter values: cloud covered (0), ice (1), specular lead (2-5), dark lead (6-9), and unclassified
(-1). Further classifications within the dark and specular categories distinguish rough v. smooth and the background photon
rate. IS2 also detects dark or gray ice that might ordinarily be recorded as ocean in passive microwave calculations (Petty et al.,
2021).

We define the linear ice fraction (LIF):

$$LIF = \frac{\text{length of ice segments}}{\text{length of all surface segments}}, \tag{1}$$

where we do not include cloud-covered (seg_surf_type = 0) or unclassified segments (seg_surf_type = -1). LIF is a one-
dimensional analog of the SIC, which can be calculated in a domain either based on a single satellite pass (using all 6 beams),
or by compiling many intersecting passes as we explore in Horvat et al. (2024). We pre-process all ATL07 IS2 tracks by
removing anomalous segments longer than 200m and eliminating segments that have fewer than two neighboring segments
within 1 km along-track as in Horvat et al. (2020).





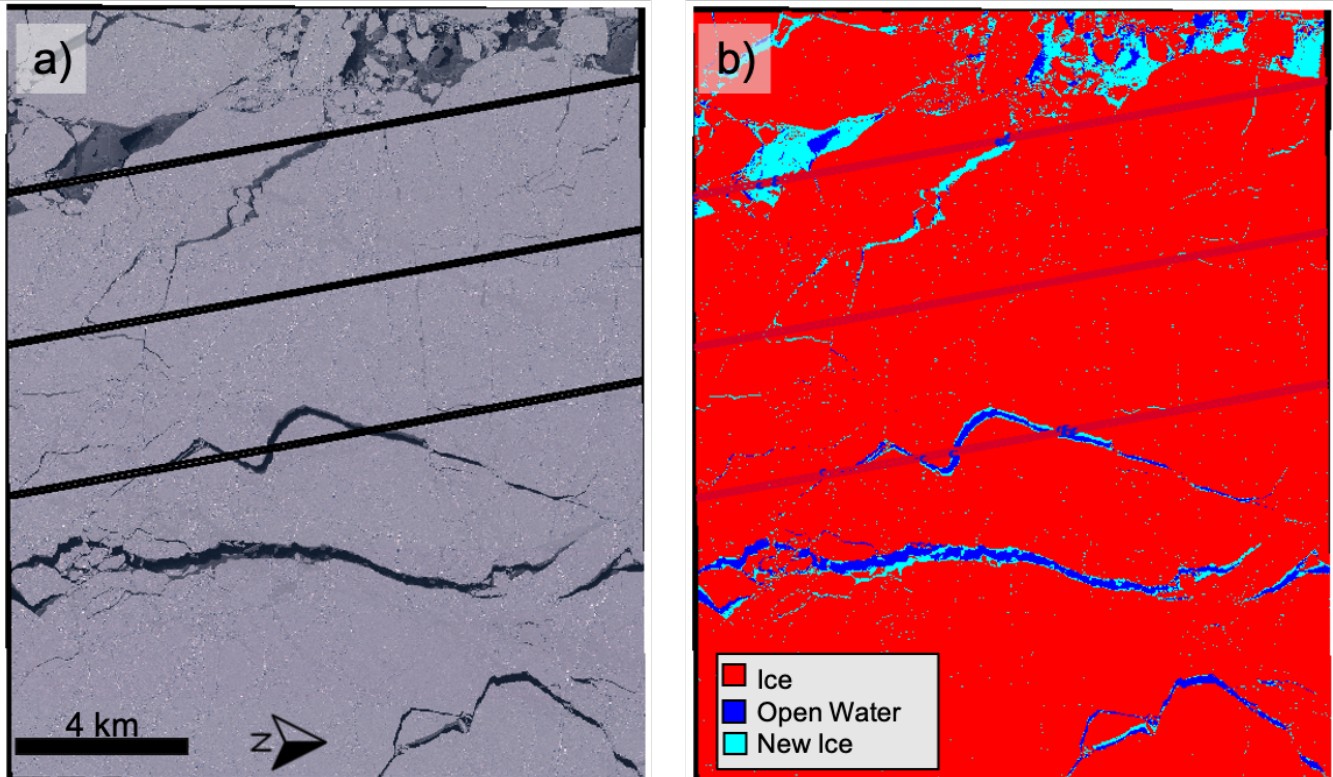

**Figure 4.** (Left) RGB Worldview-2 image taken on April 7, 2020. Straight lines are the overflight of the ICESat-2 laser altimeter. (Right) Classification of the image by the B20 algorithm into open water, new ice, and sea ice.

## 4.1 Comparison of a single-pass LIF with observations

The LIF product has promise in its ability to improve estimates of SIC, but alone it may not accurately represent a two-
dimensional field like SIC. We examine four high-resolution images coincident in space and near-coincident in time with IS2 overflights in regions with a high concentration of leads, three from Sentinel-2 optical imagery and one from the WorldView satellite (shown in Fig. 4). WorldView-2 is a member of the Maxar WorldView Legion with commercial satellites providing high resolution multispectral imagery. The red, green, and blue bands have 1.85-m resolution, higher than the IS2 footprint. The Sentinel-2 (S2) mission consists of a pair of satellites carrying the multispectral instrument (MSI) acquiring data in 13
bands. The red, green, blue, and near-infrared bands (B02, B03, B04, and B08) are at 10-m resolution, a resolution similar to the IS2 footprint (Drusch et al., 2012). We examine 25 km x 25 km areas of the Sentinel-2 imagery with the ICESat-2 tracks intersecting 25 km of the image. The ICESat-2 tracks transect the 14 km x 17 km WorldView image for 14.2 km. Following (Buckley et al., 2023), we classify the WorldView and Sentinel-2 image pixels into surface types: open water, ice, and other. The other pixels in these scenes are associated with new ice that appears gray in color which, for SIC and LIF calculations, is
considered ice.




We account for the time difference between the imagery and the ICESat-2 overpass by applying a drift correction. We use the daily sea ice motion vectors from the Polar Pathfinder Daily 25 km EASE-Grid Sea Ice Motion Vectors (Tschudi et al., 2019) to find the daily average magnitude and direction of the ice drift. We then multiply this value (in ms$^{-1}$) by the seconds between the image acquisition and the ICESat-2 sampling. The magnitude of the ice drift between the sampling ranges from

22 m over 15 minutes to 270 m over 2 hours and 8 minutes.

We calculate two different LIF metrics for each drift-corrected scene. First, we calculate a "best" LIF by extracting the value of the classified WorldView imagery at the location of each ATL07 segment for all six IS2 beams. Second, we use the values of the ATL07-derived surface type classifications after post-processing to determine the ATL07-derived linear ice fraction (Eq. 1). We compare these against the local values of optically-derived SIC from the imagery and the local PM-derived SIC (bound

to the area of the optical imagery) from the five products listed in Table 3. In each case we compute biases relative to the optical imagery SIC. Note that although the optical imagery is drift-corrected, the timing of the ICESat-2 and PM satellites are also asynchronous. However, given the resolution of PM SIC data are on the order of km, we do not drift-correct these data. Nonetheless, some biases could emerge because of changes to sea ice over that domain between satellite passes.

In Table 3, we tabulate biases for all images, and compile the average imagery bias by taking the mean of the absolute bias

across the four images. Even for a single pass, the "best" and ATL07-based IS2 LIF outperforms the PM-SIC products, with a mean bias of 1.0% and 2.4%, compared to mean biases of at least 2.9% for the PM products. This is especially notable in the May 7, 2022 image, an area of highly fractured sea ice which is considered completely ice-covered by four of the PM satellites. For all four images, the NSIDC CDR estimates 100% SIC, though the imagery shows between 1.3% and 9.5% open water fraction. The standard ATL07 product outperforms the PM products, with a median error for ATL07 classification similar

to the best-case error for all PM-SIC retrievals in OIB data (see Table 2). Still, there remains substantial room for improvement in ATL07 surface classification - a further 60% improvement above the ATL07-based LIF is possible, to a "best" bias of just 1.0%, in these imagery. This "best" bias is determined by the correlation between IS2 ground tracks and the crack features of the sea ice. Although there may be a general correlation between lead geometries and IS2 ground tracks, we show in Horvat (2024) that the expected value of this bias in the Arctic is effectively zero. Therefore, the difference between the the "best"

and the "ATL07" scenario indicate some error in either the drift correction or the ATL07 classification. Regardless, it is clear that improvements in ATL07 classification could lead to an IS2-based SIC product that improves substantially upon the error characteristics of PM-SIC data in high-concentration ice regimes.

## 5   Conclusions

In this study, we evaluated the skill of commonly used PM-SIC algorithms in representing local sea ice concentration, compared

to high-resolution optical imagery from Operation IceBridge, optical satellite sensors, and an estimate of sea ice concentration from individual passes of the IS2 laser altimeter. We showed that, in general, PM-SIC measurements have positive biases in winter conditions over compact sea ice, consistent with the existing literature (Kern et al., 2019). During the summer, we observe that all PM products exhibit a positive median bias, with the exception of the NASA Team algorithm. There is also



a greater median bias and a wider spread between the 25th and 75th percentiles in the summer. In both winter and summer,
the 25th to 75th percentile range includes both negative and positive values NASA Team, OSI SAF and ASMR-ASI products,
while the NSIDC CDR, Bootstrap, and AMSR-NT product biases are all positive from the 25th to 75th percentile. We also
find that the passive microwave bias is related to the melt pond fraction. The mechanisms of these biases are complicated and
a result of the algorithm as well as the sensor limitations. While a precise algorithmic and sensor comparison is not within the
scope of this study, it invites future work to understand why, on these subsets of data, there is such a systematic difference.

We also examined the ability of the IS2 laser altimeter to estimate sea ice concentration. We provided four examples where
IS2 passes are coincident with high resolution imagery. We validated the surface type classification parameter in IS2's sea ice
height product, ATL07, against the classified imagery and found good agreement. The single-pass linear ice fraction (LIF)
from ATL07 was on average 1.4 % greater than the true along-track sea ice concentration, and 2.4 % greater than the two-
dimensional image sea ice concentration. In these four cases, the LIF is more representative of the high resolution image
scene than all passive microwave products on this date and location. For near-100% sea ice, the IS2 altimeter can produce
comparable or improved estimates of SIC even for a single overflight of a sea-ice-covered area, and with improvements to the
ATL07 surface classification scheme has room to reduce open water biases significantly.

This paper highlights the limitations and uncertainties in the passive microwave sea ice concentration products and presents
a promising new method for estimating ice concentration and lead fraction, especially in areas of high sea ice concentration
with narrow leads. In the second part of this paper (Horvat, 2024) we develop an LIF emulator that samples optical imagery at
the frequency and direction of IS2 to understand the limitations of a one-dimensional product. We find the minimum number
of IS2 passes for an accuracte estimation of SIC in a range of ice conditions and build a monthly LIF product that has similar-
or-better error than PM data compared to classified imagery. In these two papers we demonstrate how ICESat-2 may be used
to determine sea ice concentration and lead fraction at high resolution.

*Data availability.* The classified OIB imagery is archived on Zenodo: https://doi.org/10.5281/zenodo.13129097.

*Author contributions.* EB classified the imagery and compared imagery SIC with PM SIC. CH conceived of and developed the LIF product.
PY helped with the WorldView ICESat-2 analysis. EB and CH wrote the paper. All authors consulted on the scientific approach and content.

*Competing interests.* The authors declare no competing interests.



*Acknowledgements.* CH and PY were supported by Schmidt Futures—a philanthropic initiative that seeks to improve societal outcomes
through the development of emerging science and technologies. CH acknowledges support from NASA 80NSSC20K0959, NASA 80NSSC23K0935, and NSF 2146889. EB is supported by NASA 80NSSC23K0782.



## Appendix A: Alternative Sampling Method

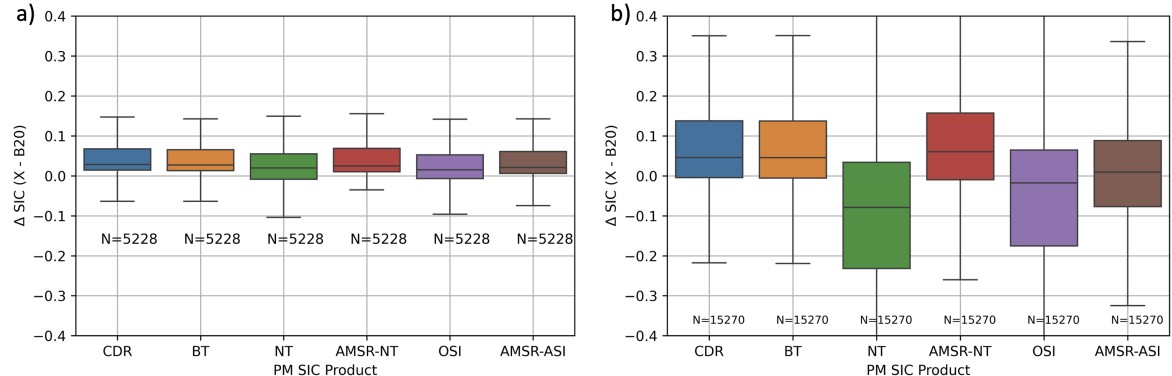

**Figure A1. Differences between Passive Microwave SIC retrievals and Operation IceBridge SICwith alternative sampling method for winter scenes (a) and summer scenes (b).** Same as in Figure 2 but individual OIB SIC is compared with the PM grid cell value in which that image falls. $\Delta$ SIC is given as the PM product less the OIB SIC value where values $> 0$ indicate the PM SIC product is great than imagery-derived SIC. Distribution of SIC biases for winter scenes where OIB SIC $\leq 99\%$. Each boxplot shows the interquartile range (IQR) which is 25th percentile to 75th percentile. The line inside the boxplot represents the median. The whiskers show the range which here is defined as 1.5 times the IQR.



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
