# Peer review of "Sea Ice Concentration Estimates from ICESat-2 Linear Ice Fraction. Part 1: Multi-sensor Comparison of Sea Ice Concentration Products"

_EGUsphere, 2024_

## Author Comment (AC1)

Summary

This paper presents a new technique for understanding fractional sea ice coverage in the Arctic, by developing a Linear Ice Fraction (LIF) product from ICESat-2 ATL07 data. It's great to see the high-resolution capabilities of ICESat-2 being used for this novel application. The paper was well-structured and enjoyable to read, and I have just a few comments to address prior to publication.

We thank the reviewer for their thoughtful and constructive comments. We appreciate the time and effort they took to read and evaluate our manuscript. The suggestions and feedback have helped us clarify and improve the presentation and interpretation of our results. Below, we provide detailed responses to each comment and describe the corresponding changes made to the manuscript.

Comments

L6-7: The statement comparing winter and summer biases is a little misleading. Without the further context provided in the paper, it reads as if summer biases are consistency smaller, rather than skewed by the NT algorithm. It would be useful to highlight here that in most cases, summer biases are larger. See also my comments on Section 3.2.

Thank you for your comment- we addressed this in Section 3.2 and have added this section in the abstract:

"Despite the low mean bias in the summer, uncertainty increases in the summer due to complex surface conditions, leading to a wider spread in SIC biases than in winter."

L12: "…measurements of the sea ice surface **with PM data** to enhance…". IS2 LIF is still dependent on PM SIC data.

IS2 LIF is not directly dependent on PM data. Indeed the IS2 ATL07 product extent is confined by the PM SIC, as the calculation of LIF can only exist in regions where PM SIC exceeds 15%, ATL07 and the LIF do not use radiometric features similar to PM brightness temperatures and are calculated independently of passive microwave retrievals. Since our focus is on leads within highly concentrated ice, the 15% constraint does not affect our analysis.

L26: Quantify "narrow", because it's an important point for justifying why LIF are useful

Added "(ranging from one meter to hundreds of meters)"

L41: I disagree with the introduction of LIF being an independent measure of sea ice presence. The LIF is developed using IS2 data that rely on a PM concentration product to determine sea ice presence, so LIF is more complimentary than independent. Please make this clear through the paper.

Please see the response to the comment above. We have clarified this point in the revised text at the end of the description of the LIF calculation:

The ATL07 product is limited to regions with passive microwave SIC > 15\%, but LIF is derived from the ICESat-2 surface type classifications that rely solely on ICESat-2's photon cloud and not surface brightness temperatures and is thus independent of passive microwave inputs, especially given the high-concentration ice we consider here.

L50: "**Then,** using…"

Added "Then,"

Table 1: This might be an EGU issue, but the date formatting in the table wasn't great to read

Reformatted the dates per egu style guidelines

Table 1, row 2, column 6: Remove "–"

Corrected with a previous comment on the data formatting

Table 1, row 5, column 6: Do you mean 450 and 430?

Yes, corrected.

L76: "…advanced **over the satellite period**…"

Added "over the satellite period"

L115: "instrument" > "instruments"

corrected

L120: "utilizing" > "utilizes"

corrected

L135: The OIB acronym hasn't been defined

Now defined at the beginning of the paragraph

L142-143: What do the authors mean by "outliers", and why this becomes more of an issue when MPF is greater than 50%?

By "outliers," we are referring to images where the very high MPF (MPF>50%) may not be representative of actual surface conditions- due to either small image footprints that are not representative of the whole PM footprint area or potential misclassifications in the image processing routine. This is only 2% of images. We have clarified this in the text.

We have edited the text in the manuscript:

In the summer, we restrict the analysis to images with MPF<= 50% to reduce the influence of potentially misclassified images that may produce unrepresentatively high melt pond fractions.

L149: Remove "(box)" ?

removed

L152: "these products" > "the PM products"

corrected

L153: Should the "(2)" say "(Figure 2a)" ?

Yes, thank you. Corrected.

L157-158: I couldn't make much sense of this sentence. What do the authors mean by "strong similarity in patterns" ?

Changed to "NSIDC biases have a mean and range similar to the BT biases."

Section 3.2: The results here are particularly interesting, and I'd like a bit more information on why PM products exhibit a positive SIC bias in summer, and why it's larger than winter. In Section 1 the authors explain that melt ponds on the sea ice appear radiometrically similar to open water, so if anything I'd expect PM to underestimate SIC compared with imagery. It would be great to add some brief text relevant to this in the abstract and Section 1 too.

This is an important point that warrants further discussion. We have updated Section 3.2:

"We find that the NT product provides the lowest SIC estimates among the algorithms evaluated, with this negative bias more pronounced in summer than winter. This is consistent with findings from Kern et al., 2020, who showed that NT products tend to underestimate SIC in the Arctic during summer due to their high sensitivity to surface melt and use of fixed, hemispheric tie points that do not capture evolving surface conditions. In contrast, we found products using the BT, NT2, and NSIDC algorithms tend to overestimate SIC, with biases of 5%–10%, consistent with Kern et al., 2020. Kern et al., 2020 also identified the OSI SAF product as having the lowest absolute bias, which aligns with our findings (Fig. 2b and Table 2).

These varying biases reflect the challenges PM algorithms face in summer when complex surface conditions—such as widespread melt ponds, wet snow, and variable ice concentrations—distort the microwave signal. While melt ponds can cause underestimation when misclassified as open water, they can also lead to overestimation when their presence affects the determination of tie points. Algorithms like OSI SAF attempt to mitigate this by using daily-updated dynamic tie points, whereas NT and NT2 rely on static tie points that are not adapted to melt season variability. These contrasting sensitivities to melt processes contribute to both under- and overestimation of SIC and explain the wider spread in PM SIC error observed in summer (Fig. 2b) compared to winter (Fig. 2a).

L168: "Figure 2b **and Table 2**"

Added "and Table 2"

L172: The NT2 acronym hasn't been defined

Now defined in the algorithms section

L172-173: Could the authors explain why they find this interesting? Because the changes to NT2 weren't intended to account for ponding.

We agree that the NT2 algorithm was not designed to account for melt ponding. We did not intend to suggest otherwise but highlight that both the NT algorithms (original and NT2) show the greatest biases at high melt pond fractions. We believe this is worth noting, as it suggests even the new algorithm remains sensitive to surface melt conditions. We removed the word "interesting" to prevent this misunderstanding.

L182: An IS2 footprint of 10 m was stated in Section 1, and 11 m here

Changed to 11 m in section 1 and reference added.

L187-188: What is meant by "likely recorded"? And what impact would this have on the IS2 products?

This sentence was reworded for clarity: "Kwok et al. (2019b) found that IS2 can consistently resolve leads as narrow as 27 m, although due to the incidence angle of ICESat-2 relative to the orientation of the lead, finer scale cracks are likely still represented in IS2 sea ice products (Hell and Horvat, 2024).

General: I suggest each author has another readthrough and checks for clarity and accuracy in the text. I noticed some issues with grammar/typos/formatting (citations and symbols).

Thank you for the suggestion, the authors have read through the manuscript again.

---

## Author Comment (AC2)

This paper provides an evaluation of passive microwave (PM) sea ice concentration (SIC) estimates using classified airborne visual imagery from NASA's Operation IceBridge (OIB) and introduces a new ICESat-2-based Linear Ice Fraction (LIF) dataset. The comparison includes classified imagery from four satellite imagery scenes (Sentinel-2 and WorldView). The study is a follow-up to a previous submission, which I also reviewed. In response to that feedback, the authors have now split the original manuscript into two parts to allow for a more focused discussion. This is a good approach, but I am somewhat uncertain about how well it has worked in practice.

We thank the reviewer for their thoughtful and constructive comments. We appreciate the time and effort they took to read and evaluate our manuscript. The suggestions and feedback have helped us clarify and improve the presentation and interpretation of our results. Below, we provide detailed responses to each comment and describe the corresponding changes made to the manuscript.

In this Part 1 study, much of the paper is dedicated to presenting classified airborne imagery from Operation IceBridge, which is used to highlight biases in PM SIC data. However, we already know from past research (e.g., Kern et al.) that PM SIC products contain biases, and the comparisons here mostly serve to confirm those findings. While it is valuable to have more insight into these biases, the OIB imagery is not actually used to assess the new ICESat-2 LIF dataset, even though LIF appears to be the main focus of the study. More than half of the paper is therefore spent reaffirming known PM SIC biases, rather than contributing directly to the LIF validation. My impression from the earlier review was that the authors planned to expand the comparisons with coincident imagery, ideally incorporating OIB data that overlaps with ICESat-2 tracks. There were indeed OIB cal/val flights in 2019 and a summer calibration campaign in 2022 that could have been utilized for this.

Thank you for your comment. Validation of the ICESat-2 surface type classification is not the purpose of this paper. We refer the reviewer to other studies that have conducted such validation and assessment (Kwok et al., 2020, Petty et al., 2021, Tilling et al., 2020). The novelty of this study is understanding PM biases with the OIB imagery and development of the LIF product from ICESat-2.

Kwok, R., Petty, A. A., Bagnardi, M., Kurtz, N. T., Cunningham, G. F., & Ivanoff, A. (2020). Refining the sea surface identification approach for determining freeboards in the ICESat-2 sea ice products. *The Cryosphere Discussions*, *2020*, 1-18.

Petty, A. A., Bagnardi, M., Kurtz, N. T., Tilling, R., Fons, S., Armitage, T., ... & Kwok, R. (2021). Assessment of ICESat-2 sea ice surface classification with Sentinel-2 imagery: Implications for freeboard and new estimates of lead and floe geometry. *Earth and Space Science*, *8*(3), e2020EA001491.

Tilling, R., Kurtz, N. T., Bagnardi, M., Petty, A. A., & Kwok, R. (2020). Detection of melt ponds on Arctic summer sea ice from ICESat-2. *Geophysical Research Letters*, *47*(23), e2020GL090644.

The second half of the paper presents the LIF analysis, where the authors introduce four Sentinel-2 and WorldView scenes to evaluate ICESat-2-derived LIF estimates. However, they only show one example in detail, and then provide a summary table, which makes the evaluation of this new dataset feel quite limited. A major advantage of working with a small number of high-resolution scenes should be the ability to explore different surface types, environmental conditions, and classification

performance in depth, but this aspect is underdeveloped. For example, it would be interesting to analyze how different ATL07 classification types influence LIF retrievals, how well the drift correction works, or how dark leads, which have known retrieval issues and are no longer included in the sea surface height retrievals (Kwok et al., 2021), affect the results. Similarly, while drift correction is applied, the fact that only four scenes are used means that manual adjustments could have been done instead. Recent studies such as Koo et al. (2023) and Liu et al. (2025) analyzed 17-18 coincident Sentinel-2 scenes and manually adjusted them, so this approach should be considered. Having at least one summer scene and scenes from other parts of the Arctic and the Southern Ocean would be highly beneficial.

Thank you for the comment. We agree that a more in depth study of how ICESat-2 responds to a variety of sea ice conditions would be beneficial. However, the purpose of this study is to demonstrate the PM biases relative to the optical imagery sea ice concentration and to suggest that we may take advantage of ICESat-2's narrow footprint to better resolve leads and reduce some uncertainty and biases of PM SIC in high ice concentration areas. The three papers mentioned above can provide information on the accuracy of the sea ice surface classification. We have added a sentence in the conclusion:

"We acknowledge that the LIF is a derived product and thus dependent on the accuracy of the ICESat-2 surface type classification."

As a point of clarification on the ATL07 product- dark leads are no longer included in the sea surface height retrieval, but are still included as leads and thus fall in the category of open water in the LIF determination.

Another aspect that could be explored in more detail is the definitional differences between PM SIC, optical imagery SIC, and ICESat-2 LIF. This is only briefly mentioned in L219, where the authors state that "new ice that appears gray in color is considered ice for SIC and LIF calculations." However, this is a significant issue that deserves more discussion. How do different ice types (open water, leads, gray ice) compare across these datasets? Addressing this would improve the interpretation of the results.

We agree that this is an important topic to discuss. We have added a paragraph to discuss this:

"An important consideration when comparing SIC estimates across different sensors is the definition of what constitutes "ice." Thin ice emits microwave radiation at levels intermediate between open water and thick, snow-covered ice, making it difficult to distinguish using standard PM SIC retrieval algorithms (Comiso and Sullivan, 1986). In high-resolution imagery, new ice is often visually distinct: it appears darker than first-year or multi-year ice, significantly brighter than open water, and with a near-infrared reflectance higher than that of melt ponds. These spectral and brightness differences make it relatively straightforward to develop algorithms that distinguish new ice from other surface types. While thin ice is generally classified as ice in ICESat-2 data (Figure 4), the radiometric properties of thin and thick ice remain challenging to distinguish. This study finds that passive microwave products generally overestimate SIC, where the potential underestimation caused

by misclassifying thin ice as open water is offset by the overestimation resulting from the inability of coarse-resolution sensors to resolve narrow leads. These opposing biases can obscure the true impact of thin ice on SIC retrievals."

Specific Comments:

L142: The authors state that they examine images where MPF ≤ 50% in summer to avoid outliers and misclassified images in the unsupervised analysis. However, wouldn't it be more informative to include scenes with high melt pond fractions? Why are these considered outliers?
This wording was also pointed out by reviewer 1. We have corrected this sentence for clarification.

By "outliers," we are referring to images where the very high MPF (MPF>50%) may not be representative of actual surface conditions- due to either small image footprints that are not representative of the whole PM footprint area or potential misclassifications in the image processing routine. This is 2% of images. We have clarified this in the text.

We have edited the text in the manuscript:

In the summer, we restrict the analysis to images with MPF<= 50% to reduce the influence of potentially misclassified images that may produce unrepresentatively high melt pond fractions.

L151: The paper states that PM SIC products on average overestimate SIC, but there is significant spread.

We changed this sentence to: "PM products overestimate SIC on average, but there is some spread."

L160: It is mentioned that PM SIC products have a bias on average, but they are highly variable. However, OSI SAF and ASR do not appear to be biased on average—can this be clarified?

They are biased on average–3.5% and 5.5 % are their average biases. See Table 2. We have added a reference to Table 2 into the text.

L218: The authors mention "other pixels" in the text, but in Figure 4, the term "new ice" is used instead. This should be consistent.

Thank you for pointing this out. In this case, the term *"other pixels"* refers specifically to *"new ice"* that appears gray in optical imagery, as noted in the manuscript. To improve clarity and maintain consistency with Figure 4, we have added a clarification indicating that these "other" pixels correspond to "new ice:"

"Following Buckley et al., (2023), we classify the WorldView and Sentinel-2 image pixels into surface types: open water, ice, and other (new ice)."

L225: How well does the drift correction work? This was not very clear in the methods section. Given that only four scenes are used, why wasn't a manual adjustment tested as an alternative?

In this study, we visually verified the alignment between the corrected ICESat-2 tracks and corresponding lead and floe features in the optical imagery (Figure 4 and 5). In the four cases presented, the drift correction consistently improved alignment and no additional manual adjustment was necessary. We have clarified this in the methods section and now note that visual inspection was used to assess correction quality. We added a clarification sentence:

We visually verified the effectiveness of the drift correction by ensuring that leads identified in the ICESat-2 surface type classifications aligned with corresponding leads in the optical imagery.

L236: The statement that the May 7, 2022 image represents an area of highly fractured sea ice that four PM SIC products classify as completely ice-covered is interesting. It would be useful to include a visual example of this to illustrate the discrepancy.

We agree! We have now added the May 7, 2022 image, showing the highly fractured sea ice along with the ICESat-2 ground track, to illustrate the discrepancy between the observed surface conditions and the PM SIC estimates as Figure 5.

Figure 3: This figure is hard to interpret, and a better way to display this data should be considered.

We have added text to better describe this figure and aid with interpretation:

Figure 3 illustrates the relationship between increasing MPF and PM-SIC bias across products. The delineated interquartile ranges emphasize that not only does the bias increase with MPF, but the variability across scenes also grows, underscoring the challenge of accurately estimating SIC under ponded conditions due to spectral variability in melt pond signatures.

ASI Data in Figure 3: Why is ASI data only shown in the 0-5% SIC interval? Shouldn't all products be included in every bin?

We added this to the Figure caption to clarify:

Note that the average melt pond fraction (MPF) within a PM grid cell is typically greater than 5%; only within the smaller 6.25 km ASI grid cells is the average OIB-derived MPF below 5% for some cells.

All Sentinel-2/WorldView scenes: Since there is space, why not show all four Sentinel-2/WorldView comparisons with ICESat-2 LIF? The Figure 4 scene also looks quite small—is this the full scene, or just a zoomed-in version?

We now include all four Sentinel-2 and WorldView scenes with the ICESat-2 tracks overlaid, as requested as Figure 5. As noted in the manuscript, the WorldView images are smaller in spatial extent than the Sentinel-2 scenes. Specifically, "The ICESat-2 tracks transect the 14 km × 17 km WorldView image for 14.2 km." Figure 4 shows the full extent of the WorldView image, not a zoomed-in subset.

Comparison with MODIS/Landsat PM Evaluations: The authors reference previous Kern et al. studies but do not clearly compare their results. How do the biases in this study compare with those found in MODIS and Landsat SIC evaluations? Are there any new insights gained from the OIB comparisons?

Thank you for this comment. We have added to our discussion of results in relation to MODIS-based comparisons in Section 3.2 where we had noticed similar bias patterns across algorithms (e.g., higher biases for Bootstrap and NT products). We added further discussion on how PM algorithms interpret summer melt surface conditions, incorporating relevant findings from the Kern et al. studies. We further clarify the comparison across studies, we have now added a sentence to the conclusion summarizing how our OIB-based results compare with evaluations:

"These findings are generally consistent with previous PM studies including comparisons with MODIS (Kern et al., 2020), Landsat (Kern et al., 2022), and ship-based observations (Kern et al., 2019). However, the OIB-based comparisons in this study reveal generally smaller absolute biases and provide new insights into how PM SIC may not capture the smallest-scale sea ice features seen in high-resolution imagery."

Methods and Results Organization: The methods and results are somewhat mixed together, which makes it harder to follow. It may be better to fully separate them, even if this shortens the results section.

We present the results and methodology alongside one another because there are two themes of this paper: first we do a comparison of the PM SIC and OIB SIC, and second we demonstrate the ability of LIF. Presenting the methodology and results for each together allows for greater clarity within each section and helps readers follow the logic of the analyses. We have reviewed the text and made small edits to improve clarity and better separate the methodology and results within each section.

References

Kern, S., Lavergne, T., Notz, D., Pedersen, L. T., Tonboe, R. T., Saldo, R., and Soerensen, A. M.: Satellite Passive Microwave Sea-Ice Concentration Data Set Intercomparison: Closed Ice and Ship-Based Observations, The Cryosphere, pp. 1–55, https://doi.org/10.5194/tc- 2019-120, 2019.

Kern, S., Lavergne, T., Notz, D., Pedersen, L. T., and Tonboe, R.: Satellite passive microwave sea-ice concentration data set inter-comparison for Arctic summer conditions, The Cryosphere, 14, 2469–2493, https://doi.org/10.5194/tc-14-2469-2020, 2020.

Koo, Y., Xie, H., Kurtz, N. T., Ackley, S. F., and Wang, W.: Sea ice surface type classification of ICESat-2 ATL07 data by using data-driven machine learning model: Ross Sea, Antarctic as an example, Remote Sensing of Environment, 296, 113726, https://doi.org/10.1016/j.rse.2023.113726, 2023.

Kwok, R., Petty, A. A., Bagnardi, M., Kurtz, N. T., Cunningham, G. F., Ivanoff, A., and Kacimi, S.: Refining the sea surface identification approach for determining freeboards in the ICESat-2 sea ice products, The Cryosphere, 15, 821–833, https://doi.org/10.5194/tc-15-821-2021, 2021.

Liu, W., Tsamados, M., Petty, A., Jin, T., Chen, W., and Stroeve, J.: Enhanced sea ice classification for ICESat-2 using combined unsupervised and supervised machine learning, Remote Sensing of Environment, 318, 114607, https://doi.org/10.1016/j.rse.2025.114607, 2025.

---

## Author Response (AR2)

**Response to Reviewers**

Thank you to both reviewers who took the time to give feedback again. We appreciate the attention to the responses and edits. Below, we have responded to the additional comments.

**Reviewer #1**

I thank the authors for replying to my first review. However, I feel that some of the major concerns I raised were not addressed in enough detail.

For me, the main issue is still about the aim of the paper and what we actually learn from the study. In the response document, the authors write:

"the purpose of this study is to demonstrate the PM biases relative to the optical imagery sea ice concentration and to suggest that we may take advantage of ICESat-2's narrow footprint to better resolve leads and reduce some uncertainty and biases of PM SIC in high ice concentration areas."

But when I look at the results, it seems that the higher-resolution ASI product has a similar level of agreement with the optical imagery as the ATL07 product. Only the more theoretical "IS2 Best" case shows clearly better performance to ASI. Since this result is used to justify a Part 2 study that produces a full basin-scale SIC estimate using ATL07, I think this connection should be explained more clearly. My impression is still that ICESat-2 data could be useful for SIC, but only if the lead classification is more reliable. I feel this concern was dismissed too easily by saying that it is outside the focus of this paper. But in my opinion, it is important for the goals you are trying to achieve.

We acknowledge that improvements to the lead classification is important around line 270:

"Still, there remains substantial room for improvement in ATL07 surface classification - a further 60% improvement above the ATL07-based LIF is possible, to a "best" bias of just 1.0%, in these imagery. This "best" bias is determined by the correlation between IS2 ground tracks and the crack features of the sea ice. Although there may be a general correlation between lead geometries and IS2 ground tracks, we show in Horvat et al. (2024) that the expected value of this bias in the Arctic is effectively zero. Therefore, the difference between the "best" and the "ATL07" scenario indicate some error in either the drift correction or the ATL07 classification. Regardless, it is clear that improvements in ATL07 classification could lead to an IS2-based SIC product that improves substantially upon the error characteristics of PM-SIC data in high-concentration ice regimes."

And we have added a sentence to acknowledge the relative accuracy of the ASI product in our examples and that LIF may not be an improvement in all scenarios.

The ASI 6.25 km resolution SIC performs similarly to ATL07 products in the four sample images (see Table 3). Although the LIF may not outperform all products in all scenarios, it provides a

new metric worthy of further consideration and comparison.

I also want to question the following point in the response:

"As a point of clarification on the ATL07 product, dark leads are no longer included in the sea surface height retrieval, but are still included as leads and thus fall in the category of open water in the LIF determination."

After reading the ICESat-2 ATBD again, it seems these dark leads are only identified by the radiometric algorithm, and the ATL07 team no longer uses them to estimate sea surface height or freeboard. So I am not sure why they are used here. Also, the ssh\_flag variable in ATL07 seems to be used to filter open water points based on height, why is that not used in this analysis? I think this could be explored, especially to see how it affects the LIF classification. This would be important to understand before moving on to the next part of the study.

That is correct that dark leads are identified through the ICESat-2 radiometric algorithm and are no longer used by the ATL07 team for sea surface height (SSH) or freeboard estimation due to known biases in their surface heights (Petty et al., 2021; ICESat-2 ATL07 Known Issues). It is not that they are not leads, but their heights are potentially biased due to the presence of clouds. Our analysis does not rely on these heights. Instead, we categorize segments as leads based on the ATL07 surface type classification, which still includes dark leads as "leads."

The ICESat-2 SSH flag is specifically intended to select the highest-fidelity specular lead returns for accurate freeboard estimation, not for comprehensive identification of all open water areas. For our purposes, calculating LIF, excluding dark leads would systematically undercount leads and bias our LIF and sea ice concentration (SIC) estimates low. Our goal is to account for all leads as classified by ICESat-2, regardless of their suitability for SSH retrieval. However, in Part 2 of this two part paper series, we create a product that includes dark leads, and a product that does not include dark leads.

**Reviewer #2**

Thank you to the authors for their thorough consideration of my previous comments. I'm very happy to see this paper being published. I just have two minor points prior to that:

L230-232: With such a strong emphasis on the IS2 LIF being an independent measurement, the authors should finish this section by discussing how they would mask their data in the absence of any SIC data from PM. This is more important now than ever.

Conclusions: I'd also like to see the above point raised in the conclusion, with some clarification of how "independent" the product can truly be over lower concentration regions.

The ICESat-2 ATL07 mask based on sea ice concentration is applied in the ATL07 algorithm to reduce errors in the reference sea surface height that arise in low sea ice concentration regions. Near the ice edge, the sea surface can be strongly influenced by wave activity. This can result in reference heights being tens of centimeters below the local mean sea surface (ATBD), which

would bias freeboard estimates. For this study, we are not interested in the ATL07 freeboard height, just the characteristics that help define the surface type classification. Without the PM SIC mask, we would need to inspect whether wave influence affects the return characteristics used to determine surface type. However, we expect that even without this mask applied in the ICESat-2 ATL07 algorithm, the LIF product would be effective in regions of low sea ice concentration.